# Prognostic Impact of Stromal Immune Infiltration before and after Neoadjuvant Chemotherapy (NAC) in Triple Negative Inflammatory Breast Cancers (TNIBC) Treated with Dose-Dense Dose-Intense NAC

**DOI:** 10.3390/cancers12092657

**Published:** 2020-09-17

**Authors:** Luca Campedel, Paul Blanc-Durand, Asker Bin Asker, Jacqueline Lehmann-Che, Caroline Cuvier, Cedric De Bazelaire, Luis Teixeira, Stephanie Becourt, Florence Ledoux, Hamid Hocini, Edwige Bourstyn, Catherine Miquel, Sophie Guillerm, Patrick Charveriat, Marc Espié, Anne De Roquancourt, Anne-Sophie Hamy, Sylvie Giacchetti

**Affiliations:** 1Breast Diseases Unit, Saint-Louis Hospital/AP-HP, F-75010 Paris, France; caroline.cuvier@aphp.fr (C.C.); luis.teixeira@aphp.fr (L.T.); stephbecourt@gmail.com (S.B.); florence.ledoux@aphp.fr (F.L.); hamid.hocini@aphp.fr (H.H.); edwige.bourstyn@aphp.fr (E.B.); patrick.charveriat@aphp.fr (P.C.); marc.espie@aphp.fr (M.E.); sylvie.giacchetti@aphp.fr (S.G.); 2Sorbonne Université, Université Pierre-et-Marie-Curie/Paris 06, F-75005 Paris, France; 3INSERM IMRB, Team 8, Université Paris-Est Créteil (U-PEC), F-94000 Créteil, France; paul.blancdurand@gmail.com; 4Department of Nuclear Medicine, Henri Mondor Hospital/AP-HP, F-94010 Créteil, France; 5Department of Anatomopatholgy, Saint-Louis Hospital/AP-HP, F-75010 Paris, France; askor999@gmail.com (A.B.A.); catherine.miquel@aphp.fr (C.M.); anne.deroquancourt@aphp.fr (A.D.R.); 6Molecular Oncology Unit, Saint-Louis Hospital/AP-HP, F-75010 Paris, France; jacqueline.lehmann-che@aphp.fr; 7HIPI INSERM U976, Université de Paris, F-75010 Paris, France; 8Department of Radiology, Saint-Louis Hospital/AP-HP, F-75010 Paris, France; cedric.de-bazelaire@aphp.fr; 9Department of Radiotherapy, Saint-Louis Hospital/AP-HP, F-75010 Paris, France; sophie.guillerm@aphp.fr; 10Department of Medical Oncology, Institut Curie, F-92210 Saint-Cloud, France; hamyannesophie@gmail.com; 11Residual Tumor & Response to Treatment Laboratory (RT2Lab), Translational Research Department, U932, Immunity and Cancer, Institute Curie, PSL Research University, 75005 Paris, France

**Keywords:** triple negative breast cancer, inflammatory breast cancer, tumor-infliltrating lymphocytes, TIL, neoadjuvant chemotherapy, immune response, lymphovascular invasion

## Abstract

**Simple Summary:**

Inflammatory breast cancers (IBC) are very aggressive especially the triple negative ones, highlighting the importance of prognostic and predictive factors determination. The evaluation of tumor infiltrating lymphocytes (TIL) is advised in breast cancers but little is known in inflammatory breast cancers, especially the TIL variation, before and after neoadjuvant chemotherapy (NAC). We reported a series of 31 triple negative IBC treated with dose-dense dose-intense NAC and investigated post NAC TIL and TILs variation. We showed that pre-NAC immune infiltration was lower than in the non-inflammatory breast cancer and that TIL increase after NAC is associated with a decrease in EFS. These results suggest that patients whose tumors have TIL enrichment after NAC are at high risk of relapse and could be candidates for adjuvant treatment after NAC.

**Abstract:**

Inflammatory breast cancers are very aggressive, and among them, triple negative breast cancer (TNBC) has the worst prognosis. While many studies have investigated the association between tumor-infiltrating lymphocytes (TIL) before neoadjuvant chemotherapy (NAC) and outcome in TNBC, the impact of post-NAC TIL and TIL variation in triple negative inflammatory breast cancer (TNIBC) outcome is unknown. Between January 2010 to December 2018, all patients with TNIBC seen at the breast disease unit (Saint-Louis Hospital) were treated with dose-dense dose-intense NAC. The main objective of the study was to determine factors associated with event-free survival (EFS), particularly pathological complete response (pCR), pre- and post-NAC TIL, delta TIL and post-NAC lymphovascular invasion (LVI). After univariate analysis, post-NAC LVI (HR 2.06; CI 1.13–3.74; *p* = 0.02), high post-NAC TIL (HR 1.81; CI 1.07–3.06; *p* = 0.03) and positive delta TIL (HR 2.20; CI 1.36–3.52; *p* = 0.001) were significantly associated with impaired EFS. After multivariate analysis, only a positive TIL variation remained negatively associated with EFS (HR 1.88; CI 1.05–3.35; *p* = 0.01). TNIBC patients treated with intensive NAC who present TIL enrichment after NAC have a high risk of relapse, which could be used as a prognostic marker in TNIBC and could help to choose adjuvant post-NAC treatment.

## 1. Introduction

Inflammatory breast cancer (IBC) accounts for approximately 2–4% of breast cancer (BC) cases in the United States and causes 7–10% of breast cancer-related deaths [1,2,3]. Despite improvements in survival, IBC prognosis remains poor with 10-year survival ranging from 37 to 55% [4]. IBC presents specific molecular or microenvironmental differences [3,5]. Several molecular signatures have been described more or less sensitively and specifically [6,7,8]. IBC’s distribution is different from the non-inflammatory breast cancers, with more *HER2*-positive (38–40%) and triple negative IBC (TNBC) (30%) and fewer patients with ER + tumors (45–50%) [9,10,11]. Triple negative IBC has the worst prognosis [10,12]. Only two series have reported the outcome of triple negative IBC (TNIBC) [10,12]. Masuda et al. [10] a very poor outcome in 139 patients with TNIBC with a pathological complete response (pCR) of 12.4%, five-year disease-free survival (DFS) and overall survival (OS) of 17.5% and 24.3%, respectively. Boudin et al. [12] reported a pCR of 28% and a five-year DFS and OS of 58% and 57%, respectively, in a small cohort of 20 TNIBC patients treated with high-dose neoadjuvant chemotherapy (NAC) with autologous hematopoietic stem cell transplantation (HDC-AHSCT). So far, it is recommended that one use the same neoadjuvant chemotherapy regimen in inflammatory breast cancers as in non-inflammatory disease, i.e., a three-week regimen of an anthracyclin–cyclophophamide combination followed by taxanes [13].

We previously compared the efficacy of two different neoadjuvant regimens: either a dose-dense and dose-intense cyclophosphamide-anthracycline (AC) regimen or a conventional sequential NAC regimen with cyclophosphamide and anthracycline, followed by taxanes (EC-T) in a cohort of 267 patients [14]. We showed that triple negative breast cancers benefitted the most from the intensified EC regimen with 76% EFS in the intensified EC vs. 57% at seven years in the standard arm [14].

Several studies have reported an association between high levels of stromal tumor-infiltrating lymphocytes (TIL) at diagnosis and better response to neoadjuvant chemotherapy in non-inflammatory TNBC [15,16,17,18,19]. The clinical significance of post-NAC TIL levels has been less extensively studied, and its prognostic value remains controversial [18,20,21,22,23,24,25]. To our knowledge, no study has investigated the value of pre- and post-paired stromal TIL in TNIBCs so far. Post-NAC lymphovascular invasion (LVI) has been identified as a strong independent factor predictive of poor survival in BC patients treated with NAC [26,27]. However, neither of these two cohorts included patients with inflammatory BCs, and the prognostic significance of this pattern remains unknown.

Here, we report a series of TNIBC from a single institution treated with dose-dense dose-intense NAC and evaluate the classical prognostic factors plus pCR, pre- and post-NAC TIL, delta TIL and post-NAC LVI associated with EFS.

## 2. Results

### 2.1. Patients and Tumor Characteristics

Overall, 31 patients with TNIBC were included in this study. Median follow-up was 43 months (8–103). Patients and tumor characteristics are described in Table 1. Briefly, 68% of the tumors were grade 3 and 87% had clinical lymph nodes (N1), histologically proven in 61% of them. In 20 patients, p53 mutation was determined in frozen biopsies, and was mutated in 85% of them (17/20). The *PI3KCA* mutation was also studied and found in 3 patients out of 20 (15%, two H1047R and one E545K). Median TIL infiltration at diagnosis was 10% (0–60) (Figure 1).

### 2.2. NAC Completion and Toxicities

Of the patients, 20 out of 31 (67%) received the SIM1 protocol and the other 10 the SIM2 protocol. In the SIM1, five patients (25%) were unable to receive all of the treatment because of toxicity, while all the patients in the SIM2 regimen received the entire treatment. Severe adverse events (grade 3/4) were observed in 47% of the cases (14/30), mostly hematologic (79%; 11/14 patients). Granulocyte colony-stimulating factors were used in primary or secondary prevention in 57% of the patients. No toxic deaths were reported. Two patients (7%) were not operated on because of progression during NAC. All other patients underwent mastectomy and axillary dissection.

### 2.3. Post-NAC Tumor Characteristics

Pathological response was assessed in all patients who underwent surgery (29/31). Out of 29 patients, 9 patients achieved pCR (31%). Out of the 22 patients who did not achieve pCR, 19 patients had lymph node involvement (86%). TIL levels decreased in 17 patients (61%), did not change in 4 patients (14%) and increased in 6 patients (21%) (Figure 2A).

Median delta TIL was −9% (−50% up to +40%) and median TIL post-NAC was lower than before NAC (1.5% vs. 10%) (Figure 2A). Median post-NAC TIL levels differed significantly between tumors with and without pCR (no pCR: 2% vs. 0% in pCR, *p* = 0.02) (Figure 2B,C).

LVI was found in 13 out of 29 surgical specimens (45%), all of which failed to reach pCR.

### 2.4. Event-Free Survival

With a median follow-up of 49 months, 16 patients had metastatic recurrences (52%) and 3 of them had cerebral metastases (19%). Median EFS and OS were not reached. The three-year EFS and OS were 54% and 68%, respectively.

Positive delta TIL, presence of LVI and absence of pCR after chemotherapy were significantly associated with a decrease of EFS in the whole population (Figure 3).

### 2.5. Univariate and Multivariate Analysis for EFS

After univariate analysis, post-NAC LVI (HR 2.06; CI 1.13–3.74; *p* = 0.02), high post-NAC TIL (HR 1.81; CI 1.07–3.06; *p* = 0.03) and positive delta TIL (HR 2.20; CI 1.36–3.52; *p* = 0.001) were significantly associated with impaired EFS (Table 2). After multivariate analysis, only a positive TIL variation remained negatively associated with EFS (HR 1.88; CI 1.05–3.35; *p* = 0.01).

## 3. Discussion

In this unique cohort of 31 TNIBC cases treated with dose-dense dose-intense chemotherapy, we show that the increase of stromal immune infiltration between baseline tumor and post-NAC is independently associated with an impaired prognosis.

In our study, the rate of pCR was 29%. Masuda et al. [10], in a monocentric cohort of 139 patients with TNIBC, reported a pCR of 12%. In the studies using HDC-AHSCT for IBC, the pCR rate was not higher—21% in the Dazzi et al. [28] study and 28% in the Boudin et al. [12] study—but the treatment was more toxic and expensive [29]. In our study, the pCR group had a significantly higher median EFS than the non-pCR group. These results confirm that pCR is a major prognostic factor and a good surrogate of EFS, as reported in non-inflammatory triple negative breast cancer in larger series and meta-analyses [10,30,31].

The number of post-NAC LVI cases was higher than in a large cohort of 330 patients with non-inflammatory TNBC (19.4% vs. 45% in our series) [26]. This difference could be explained by the population (inflammatory vs. non inflammatory) and by the fact that the slides were prospectively reviewed. The pejorative prognosis associated with the presence of LVI after chemotherapy described by Hamy et al. [26] was found in our cohort of TNIBC, but was no longer significant after multivariate analysis; this could be due to the small number of patients.

The prognostic impact of TIL before neoadjuvant treatment (pre-NAC TIL) has been extensively reported, particularly in TNBC [15,16,18,19], but was not confirmed in our study in terms of association with EFS. Notably, the pre-NAC TIL rate in our study was low. In the pooled analysis of 3771 patients treated with NAC, the authors reported that in the 906 TNBC patients, 30% had pre-NAC TIL > 60% and 31% low TIL ≤ 10% [19]. In this series 183 patients had IBC (not only TNIBC) and 50% of them had low pre-NAC TIL (< or = to 10%), and only 15% > 60%, suggesting that patients with IBC have a lower TIL rate [19].

In our study, the major prognostic factors were the TIL variation before and after NAC and the post-NAC TIL, which represent the immune reaction to chemotherapy. The TIL evaluation was also done in patients with a pCR. A TIL level increase was the only factor associated with impaired EFS in the entire population. There are more and more publications studying the TIL variation before and after NAC with different results, sometimes even opposite results [18,21,22,24,26,32] (Table 3). Dieci et al. [21] reported for the first time the impact of post-NAC TIL in 278 patients with TNBC. The authors found that high post-NAC TIL with a cut off of 60% was associated with a better OS [21]. Pelekanou et al. [22] showed, with a cohort of 58 patients whose15.5% with TNBC (9 patients), that TIL increase after NAC was associated with improved recurrence-free survival [22]. Four other studies reviewed the impact of post-NAC TIL but only two in TNBC exclusively [18,19,24,25]. Hamy et al. [25], in a series of 716 breast cancers treated with NAC, showed that high post-NAC TIL levels were associated with aggressive tumor characteristics and with impaired EFS in *HER2*-positive breast cancers but not in luminal tumors or TNBC. The authors suggest that a strong inverse correlation exists between pre-NAC TIL levels and the variation of TIL levels [25]. Castaneda et al. [18] did not find any association in 98 patients with TNBC between post-NAC TIL and outcome. Luen et al. [24], in a large series of 375 TNBC cases, analyzed the prognostic value of TIL on residual cancer burden (RCB). They showed that the median RCB TIL level was 20% and that TIL levels were significantly lower with increasing post-NAC residual tumors and nodal stages [24]. Higher RCB TILs were significantly associated with improved recurrence free outcomes, (HR 0.86; CI (95%) 0.79–0.92; *p* < 0.001) and OS (HR 0.87; CI (95%) 0.80–0.94; *p* < 0.001) and remained significant predictors in multivariate analysis. This large series showed opposite results to ours [24]. A small number of publications showed that patients with residual tumors after NAC and post-NAC high lymphocyte infiltration had worse outcomes [32].

Here we found that increased TIL after NAC is associated with a pejorative outcome. It is important to note that the median TIL is low compared to non-inflammatory TNBC (median pre-NAC TIL: 10%; median post-NAC: 1.5%) whereas in the series from Curie, median pre-NAC TIL is 28% and median post-NAC TIL is 15.4% in the TNBC [25].

The monocentric nature of the study and the rarity of the disease limited the number of patients studied, and the study was unequipped to address most of the additional analyses. In addition, the results were limited by the lack of a control arm. As such, a multi-center study with a larger number of patients and a control arm with non-triple negative IBC patients was necessary to confirm our results.

## 4. Patients and Methods

### 4.1. Patient Selection

In this prospective cohort we included all consecutive female patients over the age of 18 with a clinically inflammatory (stage T4d), histologically proven, non-metastatic primary invasive BC with a triple negative phenotype treated with NAC from January 2010 to December 2018 at the breast disease unit at Saint-Louis Hospital, France. Ethics and Patient Consent. This study was approved by the breast disease scientific board, Senopôle, Hôpital Saint Louis. Informed consent of patients was waived by this board.

### 4.2. Diagnosis

Breast cancer was diagnosed on 14G core-needle biopsies (Postolet Achieve, Merit Medical, Maastricht, The Netherlands). A lymph node biopsy was also performed if axillary lymph nodes were palpable or found at axillary ultrasound. Estrogen and progesterone status was determined by immunohistochemistry (IHC) and positivity cutoff was 10% staining [33]. HER2 determination was systematically performed by IHC with control by fluorescence in situ hybridization (FISH) or Silver-enhanced in situ hybridization (SISH) for ambiguous cases.

### 4.3. Treatment

From January 2010 to December 2018, all the patients received six cycles of a dose-dense dose-intense neoadjuvant cyclophosphamide (1200 mg/m^2^ d1) and epirubicin (75 mg/m^2^ d1) every two weeks (SIM (Sein Inflammatoire ou Métastatique) regimen [34]). Between 2010 and 2015, patients received three cycles of docetaxel (100 mg/m^2^) and cyclophosphamide (600 g/m^2^) every three weeks after surgery (SIM1). From 2016 onwards, taxanes were added sequentially after dose-dense regimen, either with four cycles of docetaxel (100 mg/m^2^) every three weeks or weekly paclitaxel (80 mg/m^2^) for 12 cycles (SIM2). No further chemotherapy was administered after surgery. In the absence of tumor progression during chemotherapy, all patients should have a mastectomy and axillary dissection. All patients received radiotherapy after surgery in the breast and lymph nodes.

### 4.4. Pathological Response

Pathological complete response (pCR) was defined as the absence of infiltrative carcinoma in the breast and in the lymph nodes; persistent lesions of carcinoma in situ of the breast was considered as a complete response [35].

### 4.5. TIL and Lymphovascular Invasion (LVI)

Pretreatment core needle biopsies and post-NAC surgical specimens were reviewed for the purpose of the study, and were evaluated independently by two anatomopathologists dedicated to breast cancer for the presence of a mononuclear cells infiltrate (including lymphocytes and plasma cells, excluding polymorphonuclear leukocytes) following the recommendations of the international TIL Working Group [36]. The TILs were studied in all the patients including patients in pCR [37]. They were evaluated in the stroma, within the border of the tumor scar, after excluding tumor zones with necrosis and artefacts, and were scored continuously as the average percentage of stromal area occupied by mononuclear cells. Delta TIL was defined as the absolute difference between post-chemotherapy and pre-chemotherapy TIL. LVI was defined as the presence of carcinoma cells within a finite endothelial-lined space (a lymphatic or blood vessel).

### 4.6. PI3KCA and AR

The mutation status of the gene *PIK3CA* (p100 catalytic unit) for hotspots p.E542K, p.E545K and p.H1047R was determined by allelic discrimination on a Lightcycler 450 (Roche diagnostics, Meylan, France) by using a set of primers and probes specific to each mutation according to Santarpia et al. [38]. The expression of *AR* was quantified by quantitative reverse transcription PCR (RTqPCR) on a Taqman 7500 (Applied Biosystems, Les Ulis, France) using the developed test *AR* (TaqMan Assays ID Hs00171172_m1, ThermoFisher Scientific, Les Ulis, France).

### 4.7. Survival Endpoints

Event-free survival (EFS) was counted from the date of biopsy to the date of distant metastases, death or the last follow-up alive with absence of metastases ascertained, whichever occurred first. The OS was defined as the time from biopsy to death.

### 4.8. Statistical Methods

Results are reported with frequency and percent for categorical data, and median and range for quantitative data. Survival curves were estimated by the Kaplan-Meier product limit estimator and compared using partial likelihood ratio tests in Cox proportional hazards models. Hazard ratios (HRs) and their associated 95% confidence intervals (CIs) were calculated using the Cox proportional hazard model. Age, menopausal status, body mass index (BMI), clinical nodal status, SBR grade, lymphovascular invasion, TIL percentage on biopsy and mastectomy and delta TIL, were included in the univariate analysis. Variables with a *p*-value for the likelihood ratio test lower than 0.15 in univariate analysis were included in the multivariate model. Backward selection was used to establish the final multivariate model. The significance threshold was 5%. Survival probabilities were estimated by the Kaplan Meier method, and survival curves were compared with log-rank tests. Analyses were performed with R software, version 3.1.2. python 3.6.

## 5. Conclusions

To the best of our knowledge this is the first time that the impact of post-NAC TIL and TIL variation have been analyzed in TNIBC. We showed that the pre-biopsy immune infiltration seemed lower than in non-inflammatory tumors as this was already reported by Denkert et al. [19]. We also showed that TIL increase after NAC is associated with a decrease in EFS.

In the patients who achieved a pCR, the immune infiltration around the scar was very low, suggesting that when the immune cells have eradicated the tumor, they move into other areas where their action is required. This finding also corroborates the fact that persistently high TIL levels after chemotherapy could be the sign of an inefficient immune response.

Altogether, our results suggest that patients whose tumors have TIL enrichment after neoadjuvant chemotherapy are at a high risk of relapse.

## Figures and Tables

**Figure 1 cancers-12-02657-f001:**
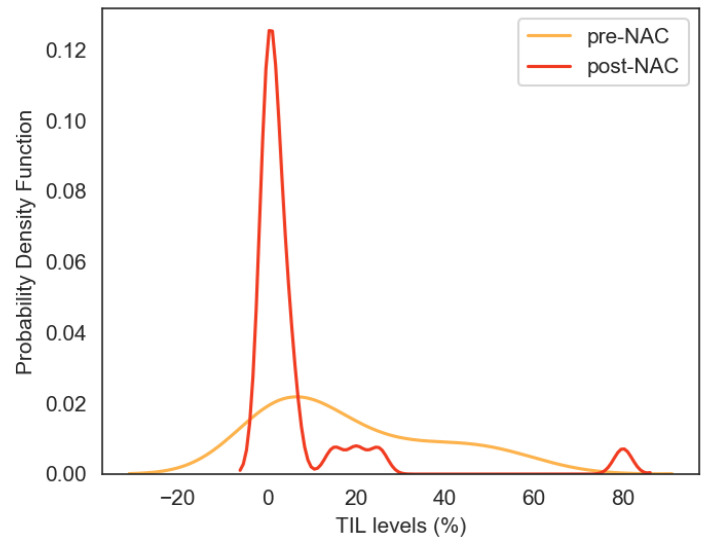
Distribution of pre- and post-neoadjuvant chemotherapy (NAC) tumor-infiltrating lymphocyte (TIL) levels (kernel density plot).

**Figure 2 cancers-12-02657-f002:**
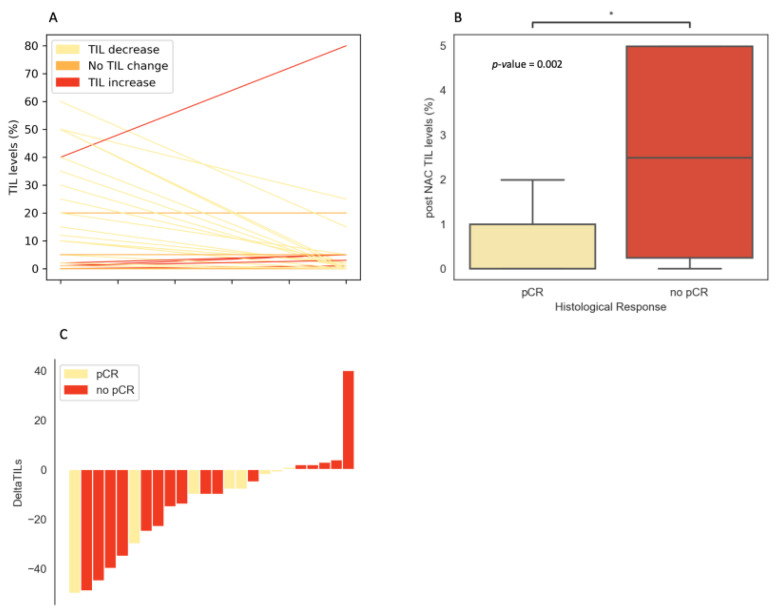
(**A**) Variation of TIL levels. Lines represent pre- and post-NAC paired TIL level values of a given patient and are colored according to TIL variation category (TIL level decrease: yellow/no change: orange/increase: red). (**B**) * Bar plots of post-NAC TIL levels according to pCR status; *p* = 0.02. (**C**) Waterfall plot representing the variation of TIL levels according to pCR status. We removed the samples with no variation from the graph.

**Figure 3 cancers-12-02657-f003:**
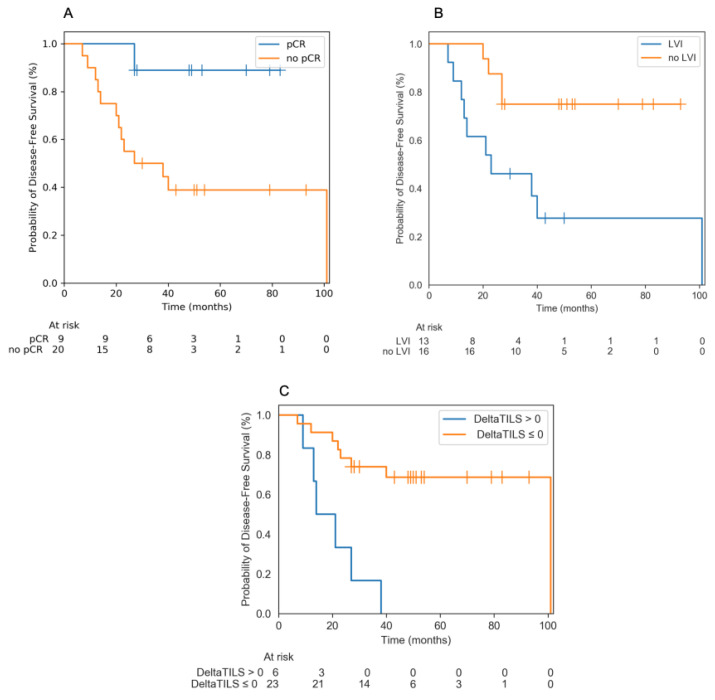
(**A**) EFS according to pCR. pCR was significantly associated with a better EFS (not reached vs. 38 months; *p* = 0.02). (**B**) EFS according to the presence of lymphovascular invasion after CNA for the entire population. Presence of LVI after chemotherapy was significantly associated with a decrease of EFS in the whole population (21 months vs. not reached; *p* = 0.009). (**C**) EFS according to ΔTILS for the entire population. TILs increase after NAC (positive delta TILS) was associated with a decrease of EFS (21 months vs. 101 months; *p* = 0.0002).

**Table 1 cancers-12-02657-t001:** Patients characteristics and tumor features.

Parameters	Number of Cases = 31 (100%)
Age at diagnosis, years	53 (28–78)
Median (range)	
Contraception, years	4 (0–30)
Menopause, yes	18 (58)
Menopausal substitutive treatment, yes	1 (6)
Body mass index, kg/m^2^	26.7 (18.4–41.1)
-<25	8 (29)
25–30	9 (32)
>30	11 (39)
Breast cancer familial history, yes	14 (45)
First degree, yes	8 (26)
BRCA, mutation	
Whole population	5 (16)
Tested population	5/11 (45)
Grade	
−1	0
−2	10 (32)
−3	21 (68)
N1/2/3	27 (87)
histologically proven	19 (61)
*P53*, mutation	17/20 (85)
*PI3KCA*, mutation	3/20 (15)
*AR*, surexpression	5/20 (25)

**Table 2 cancers-12-02657-t002:** Univariate and multivariate analysis of EFS in the whole population.

Variable	Univariate Analysis		Multivariate Analysis	
	HR (95% CI)	*p*	HR (95% CI)	*p*
Age (continuous)	1.02 (0.60–1.75)	0.94		
Menopause,				
No	1			
Yes	0.80 (0.45–1.42)	0.45		
BMI (continuous)	1.51 (0.82–2.77)	0.18		
Grade				
2	1			
3	1.10 (0.63–1.91)	0.73		
N1–3				
No	1			
Yes	1.57 (0.71–3.46)	0.26		
pCR				
No	1			
Yes	0.39 (0.15–1.01)	0.05	0.40 (0.13–1.25)	0.11
LVI				
No	1			
Yes	2.06 (1.13–3.74)	0.02	1.06 (0.50–2.27)	0.88
Pre-NAC TIL				
<median	1			
>median	0.86 (0.46–1.60)	0.64		
Post-NAC TIL (continuous)	1.81 (1.07–3.06)	0.03	1.38 (0.80–2.37)	0.25
ΔTIL				
Negative	1			
Positive	2.20 (1.36–3.52)	0.001	1.88 (1.05–3.35)	0.03

**Table 3 cancers-12-02657-t003:** Main studies evaluating the impact of post-NAC TIL on DFS/recurrence-free survival (RFS) and OS in TNBC.

Authors	Year	Number of Patients	Design	Subtype Evaluated	Correlation between Post-NAC TIL and DFS/RFS	Correlation between Post-NAC TIL and OS
Dieci et al. [21]	2014	278	Retrospective	TNBC	Positive (HR: 0.86; 95% CI 0.79–0.92; *p* < 0.001)	Positive (HR 0.86, 95% CI 0.77–0.97, *p* = 0.01)
Hamy et al. [26]	2018	330	Retrospective	TNBC	None	Not evaluated
Castaneda et al. [18]	2016	98	Retrospective	TNBC	None	None
Pelekanou et al. [22]	2017	58	Retrospective	All subtypes (15.5% TNBC)	Positive (HR = 3.9, 95% CI = 1.179–15.39; *p* = 0.02)	Not evaluated
Luen et al. [24]	2019	375	Retrospetive	TNBC	Positive (HR: 0.86; 95% CI 0.79–0.92; *p* < 0.001)	Positive (HR: 0.87; 95% CI 0.80–0.94; *p* < 0.001)
Garcia-Martinez et al. [32]	2014	26	Retrospective	TNBC	None	None

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
