# Peer review of "Prognostic Impact of Stromal Immune Infiltration before and after Neoadjuvant Chemotherapy (NAC) in Triple Negative Inflammatory Breast Cancers (TNIBC) Treated with Dose-Dense Dose-Intense NAC"

_cancers, 2020, doi:10.3390/cancers12092657_

Round 1

Reviewer 1 Report

The authors report the prognostic role of tumor infiltrating lymphocytes (TILs) changed by neoadjuvant chemotherapy (NAC) in a small group of patients with triple negative inflammatory breast cancer (TNIBC). This is a little new for TNIBC, although there are some similar works have been done in other types of breast cancers. Here are some notes for the authors:

It seems like that your results are in agreement with that of García-Martínez et al [reference #32] but are opposite to others [#21, 22, 24, 25]. Please discuss the possible reasons for this discrepancy.

Figure 3, legends for 3B and 3C are not matched to Figures. 3C, the line of negative TILs sharply drops down by 100 month? Please confirm whether all cases got relapsed in this case.

Figure 2A, colors are not easily to be distinguished between groups.

Line 89, it would be better if the details of adverse events are provided as supplemental materials.

Table 2, the positions of positive vs negative delta TILs are reversed.

Table 3, the patient number in Hamy’s study is 330. In Pelekanou’s study, the HR represents for improved RFS, which may mislead to readers with the results of Dieci et al. [21] and Luen et al. [24], all are correlated with good outcomes.

Line 81, “Pi3kinase”

Author Response

It seems like that your results are in agreement with that of García-Martínez et al [reference #32] but are opposite to others [#21, 22, 24, 25]. Please discuss the possible reasons for this discrepancy.

Answer:

The discrepancy could be mainly due to the type of tumor (TNIBC vs non inflammatory TNBC) and the chemotherapy type (dose dense dose intense vs standard chemotherapy). Persisting high TIL levels after chemotherapy could be the sign of an inefficient immune response.

Figure 3, legends for 3B and 3C are not matched to Figures. 3C, the line of negative TILs sharply drops down by 100 month? Please confirm whether all cases got relapsed in this case.

Answer:

We thank the reviewer for spotting this mistake. Figure legends have been adapted accordingly.

We confirm that the patient with the longest follow-up has relapsed at 101 months making the probability drop to 0 at 101 months.

Figure 2A, colors are not easily to be distinguished between groups

Answer:

We updated the colormaps of figure 2A to make it more readable.

Line 89, it would be better if the details of adverse events are provided as supplemental materials.

Answer:

We tried to keep this section to a minimum and have reported only serious adverse events. Indeed, we believe that adverse events should be included in the manuscript as we are developing the efficacy of dose dense dose intense treatment.

Table 2, the positions of positive vs negative delta TILs are reversed.

Answer:

Revised as requested

Table 3, the patient number in Hamy’s study is 330. In Pelekanou’s study, the HR represents for improved RFS, which may mislead to readers with the results of Dieci et al. [21] and Luen et al. [24], all are correlated with good outcomes.

Answer:

The change was made for Hamy from 320 to 330. We agree that the studies of Pelakanou, DIeci and Luen are all correlated with good outcomes.

Line 81, “Pi3kinase”

Answer:

Has been replaced PI3KCA

Reviewer 2 Report

Authors can explain the full forms of abbreviations pCR and OS and use the abbreviations thereafter.

Author Response

Authors can explain the full forms of abbreviations pCR and OS and use the abbreviations thereafter.

Answer:

Revised as requested

Reviewer 3 Report

In the manuscript titled "Prognostic Impact of Stromal Immune Infiltration before and after Neoadjuvant Chemotherapy (NAC) in Triple Negative Inflammatory Breast Cancers  (TNIBC) Treated with Dose Dense Dose Intense NAC," authors have determined prognostic significance of stromal tumor-infiltrating lymphocytes (TIL) and lymphovascular invasion before and after neoadjuvant chemotherapy in Triple negative inflammatory breast cancer. Inflammatory breast cancer accounts for only 2-3% cases, but generally it is diagnosed at later stages and its survival rate is low. Molecular characterizations have been done, but still knowledge about this type of breast cancer is limited. Under these conditions, this study holds its importance. It will be good, if authors will consider following suggestions and improve their manuscript.

  1. This paper needs a thorough editing for language. Statements are not clear.
  2. It will be good if H&E sections from biopsies showing TIL before and after neoadjuvant chemotherapy are included in the manuscript.
  3. In the Discussion section, first authors discuss their results and then find the supporting or opposing results. They are giving more importance to the results obtained by other authors. It is becoming very confusing.
  4. Why authors have included mutational analysis on the patient characteristics. Is there any significance of these mutations in inflammatory breast cancer? They have not discussed and included in the analysis?
  5.   

Author Response

  1. This paper needs a thorough editing for language. Statements are not clear.

Answer:

We have already done language editing by Professor Clauline Giacchetti, Department of Modern & Classical Languages, 602 AH University of Houston.

  1. It will be good if H&E sections from biopsies showing TIL before and after neoadjuvant chemotherapy are included in the manuscript.

Answer:

We think it is more relevant to show variations across the cohort as a whole rather than pictures of pathology slides.

  1. In the Discussion section, first authors discuss their results and then find the supporting or opposing results. They are giving more importance to the results obtained by other authors. It is becoming very confusing.

Answer:

We have integrated our results into the often inhomogeneous literature, so we are indeed obliged to detail the studies in order to make them comparable with each other and with our work.

  1. Why authors have included mutational analysis on the patient characteristics. Is there any significance of these mutations in inflammatory breast cancer? They have not discussed and included in the analysis?

 Answer:

We have included these analyses because there are little genomic data on TNIBC in the literature. However, we are unable to conduct analyses due to the amount of missing data, as genomic analysis is not routinely performed for these tumors.

Reviewer 4 Report

The authors studied in 31 triple negative inflammatory breast cancer (TNIBC) the association of pre- and post NAC tumor-infiltrating lymphocyte (TIL) with recurrence. Using Cox regression they found that TNIBC that has increased TIL post NAC treatment has higher risk of relapse.

However, the study was mainly reporting the correlation of post-TIL with EFS, and hence the paper appears to have limited findings. The authors could consider studying the association of pre-, post-, and delta TIL with respect to other parameters using Fisher exact test. For example, if TNIBC patient with LVI tends to have higher delta TIL? If p53 mutated TNIBC tends to have higher pre-NAC TIL? What is the difference for tumors features that will have increase TIL post-NAC? Are there a way we can predict relapse using only pre-TIL? Are there differences in terms of distant metastatic site in terms of pre-, post- and delta TIL since these annotations are available?

Introduction:

The authors could consider a brief description for readers who are not familiar with TNIBC, on the difference between inflammatory and non-inflammatory in pathology and molecular profile and why a different regime is needed, etc.

Further, in addition to paragraph2, how is the dose-dense dose intense NAC regimen compared to that of MD Anderson combined modality therapy?

Ueno NT, Buzdar AU, Singletary SE, Ames FC, McNeese MD, Holmes FA, Theriault RL, Strom EA, Wasaff BJ, Asmar L, Frye D, Hortobagyi GN: Combined-modality treatment of inflammatory breast carcinoma: twenty years of experience at M. D. Anderson Cancer Center. Cancer Chemother Pharmacol.

and to that of

Chevallier B, Bastit P, Graic Y, Menard JF, Dauce JP, Julien JP, Clavier B, Kunlin A, D'Anjou J: The Centre H. Becquerel studies in inflammatory non metastatic breast cancer: combined modality approach in 178 patients. Br J Cancer. 1993, 67: 594-601.

Materials and methods/results:

Detection of p53 mutation missing. The authors presented PIK3CA, AR and TP53 mutation data however, no analysis was done. Are these mutations important determinant of outcome or these are the defining mutation in TNIBC? If the authors have no intention to study these genes I’ll suggest the authors to remove the data.

Presentation:

  • Fig. 2, the authors can consider using a paired pre- and post-NAC dot plot and indicate which patients are pCR and LVI+. The current Fig. 2a and Fig. 2b are not representative of the available data.
  • Fig. 3, Kaplan-Meier analysis should be presented for pre-NAC TIL and post-NAC TIL for comparison purpose.
  • Typo. Table 2, LVI, pre-NAC TIL, etc, the ‘,’ should be ‘.’ In p-value column. Table 2 caption, “OF” should be lower case.
  • Pg. 7 line 182, “pre-operative”
  • For Table 3, why are Hamy et al , Castaneda et al. and Garcia-Martinez et al. in the table as they have no reported outcome result? shouldn’t the content of table 3 be described in introduction instead?

Author Response

However, the study was mainly reporting the correlation of post-TIL with EFS, and hence the paper appears to have limited findings. The authors could consider studying the association of pre-, post-, and delta TIL with respect to other parameters using Fisher exact test. For example, if TNIBC patient with LVI tends to have higher delta TIL? If p53 mutated TNIBC tends to have higher pre-NAC TIL?

Answer:

We didn’t report the correlation of Delta TIL with the other clinical parameters, because  a lot of our variables are binary (such as LVI, p53 status) and theses correlations are not statiscally appropriated. We chose to study their impact on both EFS and OS. We, thus  used univariate and multivariate cox analysis to study the additional risk of events of variables independently from one to another.

What is the difference for tumors features that will have increase TIL post-NAC?

Answer:

We find the reviewer's remark very interesting and we think we will have to address this issue in the future.  Nevertheless, the small size of our cohort does not allow us to provide this information in a meaningful way.

Are there a way we can predict relapse using only pre-TIL? Are there differences in terms of distant metastatic site in terms of pre-, post- and delta TIL since these annotations are available?

Answer:

In our study, pre-TIL didn’t have an impact on EFS. We also do not have the power to study the relapse site according to pre, post, and delta TIL as only 16 patients progressed with metastases.

Introduction:

The authors could consider a brief description for readers who are not familiar with TNIBC, on the difference between inflammatory and non-inflammatory in pathology and molecular profile and why a different regime is needed, etc.

Answer:

These considerations have been developed in the introduction.

Further, in addition to paragraph2, how is the dose-dense dose intense NAC regimen compared to that of MD Anderson combined modality therapy?

Ueno NT, Buzdar AU, Singletary SE, Ames FC, McNeese MD, Holmes FA, Theriault RL, Strom EA, Wasaff BJ, Asmar L, Frye D, Hortobagyi GN: Combined-modality treatment of inflammatory breast carcinoma: twenty years of experience at M. D. Anderson Cancer Center. Cancer Chemother Pharmacol.

and to that of

Chevallier B, Bastit P, Graic Y, Menard JF, Dauce JP, Julien JP, Clavier B, Kunlin A, D'Anjou J: The Centre H. Becquerel studies in inflammatory non metastatic breast cancer: combined modality approach in 178 patients. Br J Cancer. 1993, 67: 594-601.

Answer:

The differences between the SIM protocol, the Chevallier et al. protocol and the MD Anderson’s one are the cytotoxic drugs used, the administration schedule and the dosage.

Materials and methods/results:

Detection of p53 mutation missing. The authors presented PIK3CA, AR and TP53 mutation data however, no analysis was done. Are these mutations important determinant of outcome or these are the defining mutation in TNIBC? If the authors have no intention to study these genes I’ll suggest the authors to remove the data.

Answer:

We agree that we did not include these mutation in our analysis as it was either a few number of patients (3 mutated PI3KCA out of 20 and 5 out of 20 AR mutated ) or the majority of the patients (17 out 20 of P53 mutated patients). We kept these data although we did not analyze them we felt that it is important to describe the genomic features of this rare population.

Presentation:

  • Fig. 2, the authors can consider using a paired pre- and post-NAC dot plot and indicate which patients are pCR and LVI+. The current Fig. 2a and Fig. 2b are not representative of the available data.

Answer:

We have studied as the reviewer suggests pre and pre-post NAC paired dot plot for both LVI and pCR and added them to our response. See figures 2D and 2E.

Nevertheless as no trend could be observed we decided not to include them in the revised manuscript.

  • Fig. 3, Kaplan-Meier analysis should be presented for pre-NAC TIL and post-NAC TIL for comparison purpose.

Answer:

We have added as the reviewer suggests the Kaplan Meier curve for pre and post NAC TIL. See figures 3D and 3E. We didn’t include them in the original manuscript as we thought it would add to much complexity.  

Round 2

Reviewer 3 Report

The authors have replied all the queries and manuscript can be accepted.

Author Response

We would like to thank the reviewer for his comments that helped improve the quality of the manuscript.

Reviewer 4 Report

The manuscript is largely the same despite being asked a major revision.

The authors are right that the study is underpower to address most of the additional analyses. however, this study in itself with only 31 sample and 16 events, is underpower. The authors should at least mention this as a caveat in their discussion.

The paper reported limited findings that delta TIL correlate event-free survival, and critically there is no validation cohort; there is no non-TNBC inflammatory breast cancer to show that if this is a unique feature of TNBC inflammatory breast cancer.

Why is that the patients with no pCR did better than patients who had pCR (Fig. 3A)? are the follow-up treatment different (with or without surgery)? or are those patients who have no pCR are of lower grade tumor? 

Author Response

R3

We would like to thank the reviewer for his remarks which helped to increase the quality of the manuscript.

R4

We thank the reviewer for his interest in our work and the quality of his review. We have integrated some of the suggestions, and tried to answer his questions, and explain why some of them seemed less relevant to integrate into the manuscript. We are quite willing to provide answers if he still has questions or points that he considers unclear.

1) The authors are right that the study is underpower to address most of the additional analyses. however, this study in itself with only 31 sample and 16 events, is underpower. The authors should at least mention this as a caveat in their discussion.

We included this point in the discussion (l. 189).

2) The paper reported limited findings that delta TIL correlate event-free survival, and critically there is no validation cohort; there is no non-TNBC inflammatory breast cancer to show that if this is a unique feature of TNBC inflammatory breast cancer.

Indeed it is an interesting point to know if this characteristic is limited to the triple negative subtype of inflammatory breast cancers. We did not focus on this at this stage, but preferred to report the results of our TNIBC cohort and to integrate it into the results available in the literature. We include this point in the discussion (l. 189).

3) Why is that the patients with no pCR did better than patients who had pCR (Fig. 3A)? are the follow-up treatment different (with or without surgery)? or are those patients who have no pCR are of lower grade tumor?

Thanks to the reviewer for his vigilance, this is an error of legend, the group of patients with pCR has a better EFS than patients without pCR. The correction was made in the manuscript.

Round 3

Reviewer 4 Report

no further comments.

Author Response

We would like to thank the reviewer for his remarks which helped to increase the quality of the manuscript.